# Gene expression plasticity followed by genetic change during colonization in a high-elevation environment

**Huishang She[1], Yan Hao[1], Gang Song[1], Xu Luo[2], Fumin Lei[1,3,4], Weiwei Zhai[1,3,4], Yanhua Qu[1,3]***

[1]Key Laboratory of Zoological Systematics and Evolution, Institute of Zoology, Chinese Academy of Sciences, Beijing, China; [2]Faculty of Biodiversity and Conservation, Southwest Forestry University, Kunming, China; [3]College of Life Sciences, University of Chinese Academy of Sciences, Beijing, China; [4]Center for Excellence in Animal Evolution and Genetics, Chinese Academy of Sciences, Kunming, China

**Abstract** Phenotypic plasticity facilitates organismal invasion of novel environments, and the resultant phenotypic change may later be modified by genetic change, so called 'plasticity first.' Herein, we quantify gene expression plasticity and regulatory adaptation in a wild bird (Eurasian Tree Sparrow) from its original lowland (ancestral stage), experimentally implemented hypoxia acclimation (plastic stage), and colonized highland (colonized stage). Using a group of co-expressed genes from the cardiac and flight muscles, respectively, we demonstrate that gene expression plasticity to hypoxia tolerance is more often reversed than reinforced at the colonized stage. By correlating gene expression change with muscle phenotypes, we show that colonized tree sparrows reduce maladaptive plasticity that largely associated with decreased hypoxia tolerance. Conversely, adaptive plasticity that is congruent with increased hypoxia tolerance is often reinforced in the colonized tree sparrows. Genes displaying large levels of reinforcement or reversion plasticity (i.e. 200% of original level) show greater genetic divergence between ancestral and colonized populations. Overall, our work demonstrates that gene expression plasticity at the initial stage of high-elevation colonization can be reversed or reinforced through selection-driven adaptive modification.

## eLife assessment

This study provides **useful** information on the evolution of gene expression levels and plasticity in tissues impacted by hypoxia during colonization of a high-altitude environment. Unfortunately, the evidence for the conclusions is **incomplete** because of the low sample size available.

## Introduction

Understanding the interactions between organisms and their environments is becoming increasingly important as environmental change and human activity change the original distribution of many species (*Visser, 2008*; *Wingfield et al., 2011*; *McCairns et al., 2016*; *Oostra et al., 2018*). While some organisms can easily cope with environmental change, others run the risk of local or global extinction. An organismal response to environmental change partially depends on its ability of phenotypic change (*Pfennig et al., 2010*; *Scoville and Pfrender, 2010*; *Murren et al., 2015*; *Fox et al., 2019*). For organisms that have recently colonized a new environment, phenotypic change may involve two stages. At the early stage, species may rapidly change their phenotype through plasticity without

***For correspondence:**
quyh@ioz.ac.cn

**Competing interest:** The authors declare that no competing interests exist.

involving genetic changes. Under a persisting strong selective pressure, selection may bring genetic change to canalize phenotypic changes. Consequently, phenotypic plasticity has been considered to play a stepping stone for genetically based evolutionary change at a late stage, so called 'plasticity first' hypothesis (i.e. *West-Eberhard, 2003*; *Schwander and Leimar, 2011*; *Levis and Pfennig, 2016*; *Corl et al., 2018*).

Under this hypothesis, plasticity may serve adaptive evolution in two alternative ways (*Ho and Zhang, 2018*; *Ho et al., 2020*). On the one hand, if plasticity changes the phenotype in the same direction as adaptive evolution does, genetic variation can strengthen the change toward an optimal phenotypic value in the new environment (reinforcement plasticity). On the other hand, if the plastic change works in the opposite direction as that driven by adaptive evolution, the subsequent genetic change would have to revert the initial plastic response (reversion/reduction plasticity). As previous studies on morphological and physiological traits have revealed mixed evidence for the reinforcement and the reversion of plasticity (*Pigliucci et al., 2006*; *Lande, 2009*; *Pfennig et al., 2010*; *Moczek et al., 2011*; *Levis and Pfennig, 2016*; *Corl et al., 2018*), it remains unclear if and how genetic change acts to reinforce or reverse plasticity in the wild species (i.e. *Campbell-Staton et al., 2021*). The potential challenges are how to quantify phenotypic change relative to plasticity and adaptive evolution, and how to define what genes are involved in the phenotypic change, especially as many phenotypes are likely polygenically mediated (*Novembre and Barton, 2018*; *Pritchard et al., 2010*).

Comparison of gene expression provides a potential approach to quantify the relative contributions of phenotypic plasticity and genetic variation since gene expression can bridge an organism's genotype to its cellular biology and, by extension, higher-order physiological processes (*Wray et al., 2003*; *Morris et al., 2014*). While the entire transcriptional program is orchestrated by the gene regulatory network, it is possible to trace the genes that are co-expressed or underpin the phenotypic change (*Fukao et al., 2011*). In addition, gene expression evolves in a stabilizing manner, where regulatory elements accumulate mutations that keep gene expression at an optimal level for physiological functions (*Coolon et al., 2014*; *Gilad et al., 2006*; *Hodgins-Davis et al., 2015*). Herein, we used gene expression data to quantify regulatory plasticity and adaptation, as well as the relevant genetic changes in a group of wild birds that have recently colonized a highland environment, the Eurasian Tree Sparrows, *Passer montanus* (hereafter referred to as tree sparrows).

The tree sparrow is a human commensal that has spread over a wide variety of habitats in the Eurasian continent, and that has been successfully introduced to Australia and North America (*Widmann, 1889*; *Graham et al., 2011*). This species colonized the Qinghai-Tibet plateau approximately 2600 years ago (*Qu et al., 2020*), possibly concurrently with the introduction of barley agriculture in the highlands. The Qinghai-Tibet plateau is a harsh environment with an average elevation of 4500 m above sea level (m a.s.l.) where only a few endemic animals can survive. For mammals and birds living at high elevations, hypoxia is one of the strongest selective pressures that drive physiological changes (*Barve et al., 2016*; *McClelland and Scott, 2019*). Compared to the lowland ancestral population, highland tree sparrows have evolved physiological and gene expression changes for hypoxia tolerance (*Sun et al., 2016*; *Qu et al., 2020*), but how these changes relate to plasticity and adaptive evolution is largely unknown.

In order to explore this, we acclimated a group of lowland birds to an experimentally implemented hypoxic condition similar to that where we collected highland birds (i.e. 3200 m a.s.l). We could assess gene expression plasticity and regulatory adaptation in hypoxia tolerance by studying tree sparrows collected (a) in their original lowland environment (representing the physiologically ancestral stage), (b) shortly after their exposure to the new environmental stimulus (representing the ancestral plasticity at the initially plastic stage), and (c) after having adapted to the highland environment (the colonized stage). Using the framework described in *Figure 1*, we quantified gene expression change stemming from hypoxia acclimation and highland colonization and defined genes with the expression plasticity being reinforced or reversed at the colonization stage. We test whether these genes increase genetic variation between ancestral and colonized populations using a permutation test. We use this framework in four datasets including groups of co-expressed genes associated with hypoxia tolerance and those correlated to muscle phenotypes for the cardiac and flight muscles, respectively. These independent comparisons congruently show that gene expression at the plastic stage is more often reversed than reinforced at the colonized stage, and the selection-driven genetic change depends on the magnitude of reinforcement and/or reversion plasticity.

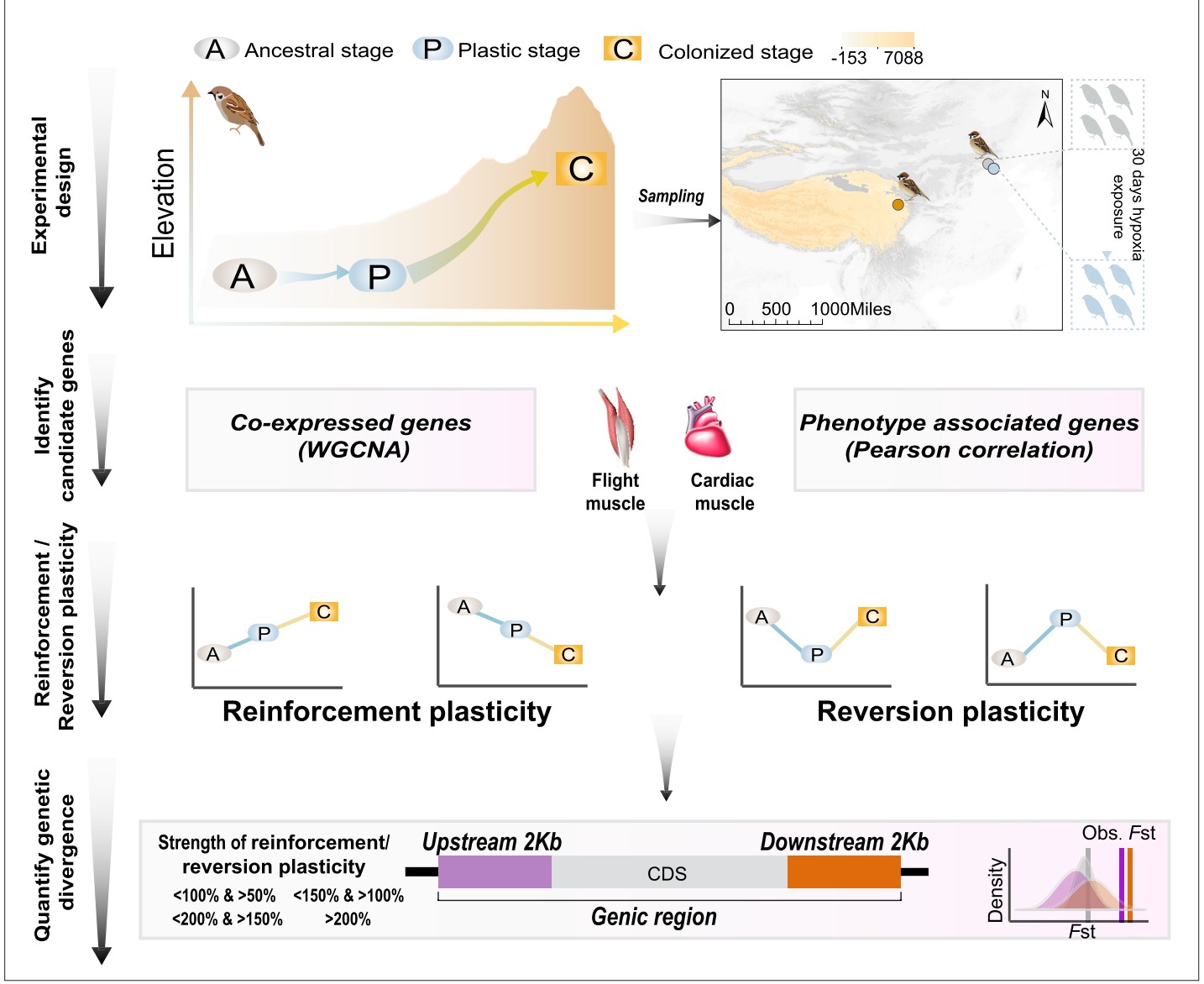

**Figure 1.** A schematic representation of quantifying gene expression plasticity and genetic change in relevant genes. Experimental design, transcriptomic data of the flight, and cardiac muscle were collected from tree sparrows obtained in their original lowland habitat (ancestral stage), experimental exposure to the hypoxic condition (plastic stage), and colonization to a high-elevation environment (colonized stage). Identify candidate genes, co-expressed genes that display correlation with hypoxia tolerance were identified by a weighted gene correlation network analysis (WGCNA) and muscle phenotype-associated genes were identified by correlating gene expression levels and each of the muscle phenotypes. Reinforcement/ reversion plasticity, the genes with reinforcement or reversion plasticity were defined by comparing expression changes between the ancestral and plastic stages (plastic change), and between plastic and colonized stages (evolved change). The strength of reinforcement/reversion plasticity is measured by a range of thresholds, i.e., <100% and >50%, <150% and >100%,<200% and >150%, and >200%. Quantify genetic divergence, genetic divergence (i.e. $F_{ST}$) between lowland and highland populations was estimated for the genic region, 2 kb upstream and downstream regions of the candidate genes. The empirical $F_{ST}$ values were compared with permuted $F_{ST}$ distributions generated from 100 random samplings from the genetic background. A threshold of $p<0.05$ was used to determine statistical significance.

## Results

### Co-expressed genes involved in hypoxia tolerance generally reverse gene expression plasticity

We sequenced transcriptomic data of the cardiac and flight muscles from the lowland tree sparrows that had been kept under the hypoxia acclimation in this study (n=7), and re-analyzed transcriptomic data in the lowland and highland tree sparrows collected by *Qu et al., 2020* (n=14, *Supplementary*

*file 1*). We focused on the flight and cardiac muscles because they affect thermogenic capacity and oxygen transport and delivery and are thus critical for the highland adaptation (*Scott et al., 2015*; *Storz et al., 2010*). Considering that response to environmental change often involves complex coordinated biological response, such as the co-regulation of genes underpinning physiological functions, we analyzed the resultant expression data for all individuals using weighted gene correlation network analysis (WGCNA, *Langfelder and Horvath, 2008*). Weighted correlation network is a system biology method for identifying groups of highly co-expressed genes (modules), summarizing module-level expression, and identifying important genes within modules (hub genes). Consequently, these co-expressed genes can capture expression-stage correlations due to the hypoxia-dependent expression. We identified six and nine modules of co-expressed genes with their expression changes that are significantly associated with the three stages (i.e. ancestral, plastic, and colonized stages) for the flight and cardiac muscle, respectively (*Figure 2A* and *Figure 2—figure supplement 1*). Across these modules, we identified 2413 and 508 hub genes based on the gene significance function (GS>first quartile and p<0.05) for the flight and cardiac muscles, respectively (*Figure 2B–C*).

Using these candidate genes we examined how gene expression plasticity changed across the three stages. We compared the gene expression levels between the birds at the ancestral stage and those at the plastic stage, as well as between the birds at the ancestral stage and those at the colonized stage. We attributed genes showing the same direction of change to the reinforcement group and those showing the opposite direction of change to the reversion group as described in *Ho and Zhang, 2018*. Using a threshold of >50% increase/reduction of gene expression between two comparisons, we attributed 36 and 15 genes to the reinforcement group and 209 and 60 genes to the reversion group for flight and cardiac muscle, respectively. Interestingly, we found fractions of genes with reversion plasticity are larger than those of genes with reinforcement plasticity (*Figure 2D*). To confirm the robustness of this observation with respect to random sampling errors, we carried out a parametric bootstrap procedure as described by *Ho and Zhang, 2019*, which aimed to identify genes resulting from genuine differences rather than random sampling errors. Bootstrap results also confirmed that genes exhibiting reversing plasticity significantly outnumber those exhibiting reinforcing plasticity (*Figure 2D*).

We then explore whether this result is robust with different magnitudes of plasticity by using a range of thresholds of <150% and >100%, <200% and >150%, and >200% to attribute genes to reinforcement and reversion groups. These different categories consistently showed an excess of genes with reversion plasticity (six out of eight comparisons, two-tailed binomial test, p<0.05, *Figure 2E–F*). To further confirm the robustness of the discovered pattern against the arbitrariness of threshold selection, we also adopted different categorizations according to the magnitude of plasticity (i.e. 20%, 40%, and 60% bin settings along the spectrum of the reinforcement/reversion plasticity), and these analyses showed similar results (*Figure 2—figure supplement 2*). Altogether, these results suggest that high-elevation colonization of tree sparrows generally reverses gene expression plasticity.

If the reinforcement and/or reversion gene expression plasticity in hypoxic tolerance were the target of selection in the high-elevation environment, we would expect to see an increase in the genetic divergence between the ancestral and colonized populations. To evaluate this we searched for genomic signatures of selection using resequencing data from 12 lowland and 11 highland tree sparrows generated from a previous study (*Qu et al., 2020*). Significantly elevated genetic divergence (i.e. $F_{ST}$) within the candidate genes beyond the genic background would provide evidence for selection acting on this group of co-expressed genes (i.e. polygenic adaptation). We thus calculated the average $F_{ST}$ for the genes with reinforcement or reversion plasticity using all SNPs included in the genic region, 2 kb upstream and downstream regions, respectively. We considered the upstream and downstream regions because gene regulation may refer to genetic variation in nearby non-coding regions. By comparing to the $F_{ST}$ distributions generated from 100 random samplings of the same number of SNPs from the genic background (see Methods), we found that the empirical $F_{ST}$ values significantly increased in the genes with the levels of reinforcement/reversion plasticity reaching above 200% (p<0.05, *Figure 2E–F*). In all cases, the empirical $F_{ST}$ values increased significantly in the 2 kb upstream and/or downstream regions of the candidate genes, but not in the genic regions. This observation thus suggests that selection-driven genetic change is more targeted on the non-coding regions and depends on the magnitude of reinforcement/reversion plasticity.

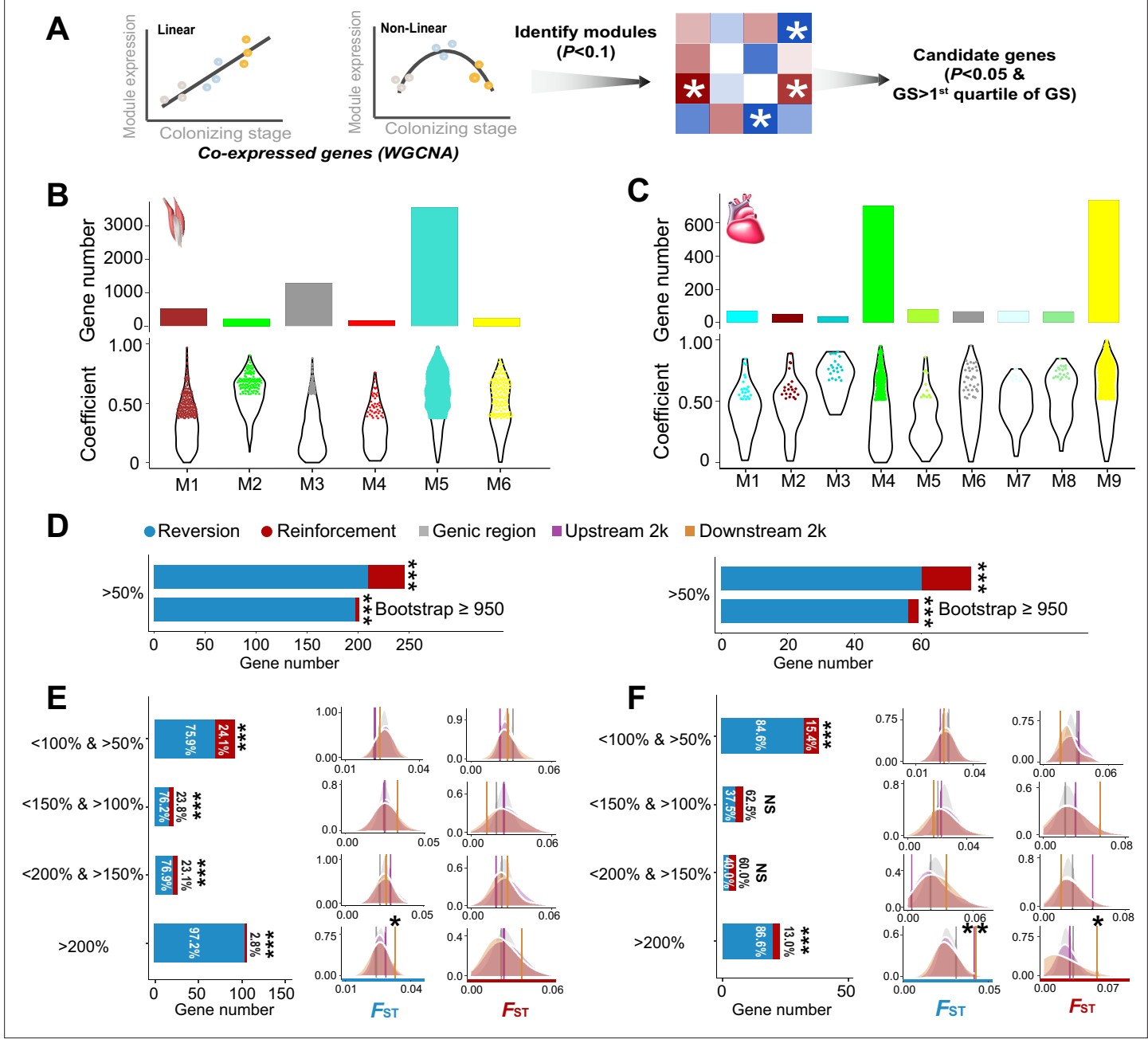

**Figure 2.** The weighted gene correlation network analysis (WGCNA) identified co-expressed genes and reinforcement and reversion of expression plasticity. (**A**) Pearson linear correlation is used to identify genes with expression reinforcement and nonlinear regression to identify genes with expression reversion. Regulatory modules were identified as branches of the resulting cluster tree via the dynamic tree-cutting method and highly correlated modules (p<0.1) were merged. We used 'GS>first quartile of GS' and 'p<0.05' to identify important genes within modules (hub genes). (**B**) and (**C**) Six and nine modules, respectively, were identified to be associated with different stages for the flight and cardiac muscles (p<0.01). Within each module, colored dots show the genes with the expression levels significantly associated with the stages (gene significance function>first quartile and p<0.05). (**D**) Frequencies of genes with reinforcement and reversion plasticity (>50%) and their subsets that acquire strong support in the parametric bootstrap analyses (≥950/1000). (**E**) and (**F**) Left, genes with expression plasticity being reinforced (red) or reversed (blue) at the colonized stage identified for the flight (**E**) and cardiac muscles (**F**), respectively. There are more genes showing reversion plasticity than those showing reinforcement plasticity. Two-tailed binomial test, NS, nonsignificant; ***p<0.001. Right, the $F_{ST}$ values (vertical lines) significantly increase in the 2 kb upstream and/ or downstream regions of the genes having the magnitude of reinforcement/reversion plasticity >200%. Vertical lines, the empirical $F_{ST}$ values; shades, permuted $F_{ST}$ distributions generated from 100 random samplings. NS, non-significant, *p<0.05; **p<0.01.

The online version of this article includes the following source data and figure supplement(s) for figure 2:

**Source data 1.** Raw data for *Figure 2B–F*.

*Figure 2 continued on next page*

*Figure 2 continued*

**Figure supplement 1.** Weighted gene correlation network analysis (WGCNA) identified co-expressed genes with gene expression change association with the three stages (i.e. ancestral stage, plastic stage, and colonized stage).

**Figure supplement 2.** Frequencies of genes with reinforcement and reversion plasticity in the flight and cardiac muscles.

**Figure supplement 3.** The median expression levels of the conserved genes (i.e. coefficient of variance ≤0.3 and average transcript per million (TPM) ≥1 for each sample) did not differ among the lowland, hypoxia-exposed lowland and highland tree sparrows (Wilcoxon signed-rank test, $p < 0.05$).

## Gene expression and muscle phenotype analyses show adaptive plasticity and maladaptive plasticity

The evolutionary outcome of the phenotypic plasticity for a population colonizing a novel environment depends on the direction of plastic change with respect to the local optimum of the novel environment. If the plastic change is close to the local optimum, i.e., adaptive plasticity, natural selection likely reinforces adaptive plasticity. Conversely, if the plasticity moves the phenotypic change away from the local optimum, i.e., maladaptive plasticity, natural selection should reduce or reverse the reaction norm and restore the phenotypic change back to the original ancestral values (*Campbell-Staton et al., 2021*). In order to investigate phenotypic plasticity and its evolutionary consequence, we analyzed two datasets that included gene expression and muscle phenotypes collected from the lowland ancestral and highland colonized tree sparrows (*Qu et al., 2020*).

Based on the correlation analysis between gene expression and muscle phenotypic values of the ancestral and colonized tree sparrows, we categorized the direction of expression-phenotype correlation as positive or negative regulators for the muscle phenotype-associated genes ($p < 0.05$, *Figure 3A*). We identified 2037 and 1866 muscle phenotype-associated genes in the flight and cardiac muscle, respectively (*Figure 3B–C*). Out of these genes, we found 328 positive regulators and 1709 negative regulators for the flight muscle, and 843 positive regulators and 1023 negative regulators for the cardiac muscle (*Figure 3—figure supplement 1*). We then characterized the differences in gene expression observed for the ancestral-plastic groups and the ancestral-colonized groups as congruent or incongruent with increased hypoxia tolerance (see Methods). Because the muscle phenotypic changes observed in the colonized tree sparrows have been shown to enhance oxygen delivery and improve metabolic capacity in the highland animals (*Giordano, 2005*; *Storz et al., 2010*; *Scott, 2011*; *Scott et al., 2015*), we deemed that the gene expressions in which the direction of plastic change matched the expectation for increase hypoxia tolerance as putatively adaptive, while those that opposed this expectation were considered to be putatively maladaptive (*Figure 3A*). Using a threshold of >50% of plasticity change, we found that the tree sparrows displayed more genes with maladaptive plasticity than genes with adaptive plasticity (two-tailed binomial test, flight muscle, 198 vs 48, $p < 2.2e^{-16}$, cardiac muscle 186 vs 166, $p < 0.05$). These results are robust with random sampling errors as our parametric bootstrap analyses also revealed similar results (*Figure 3D*).

We subsequently examined the expression-stage relationship using the thresholds (<150% and >100%, <200% and >150%, and >200%) and bin settings (20%, 40%, and 60% bin settings) as abovementioned to explore the magnitudes of adaptive and maladaptive plasticity. We found that the fractions of genes with maladaptive plasticity were larger than those of genes with adaptive plasticity (two-tailed binomial test, $p < 0.01$, *Figure 3D*). This observation is robust with different thresholds and bin settings for the flight muscle, but slightly sensitive to bin settings for the cardiac muscle, indicated by more than 50% of comparisons supporting an excess of genes with maladaptive plasticity (*Figure 3—figure supplement 2*). These results suggest that gene expression plasticity induced by hypoxia exposure at the early stage may be mainly maladaptive for muscle physiological performance in highland environment.

The observed adaptive and maladaptive gene expressions in the tree sparrows suggest that selection may have acted to reinforce and/or reverse gene expression plasticity. We then quantified the genetic divergence of these genes between ancestral and colonized tree sparrows by testing for a statistically significant increase of $F_{ST}$ as compared to the $F_{ST}$ distributions generated from 100 random samplings (see Methods). We found that the empirical $F_{ST}$ values were significantly larger in the genes within the <200% and >150% category (cardiac muscle, $p < 0.05$) and >200% category (flight and cardiac muscle, $p < 0.05$), but not in those within other categories with low levels of reinforcement and reversion plasticity. Additionally, we found that genetic divergence between the ancestral and

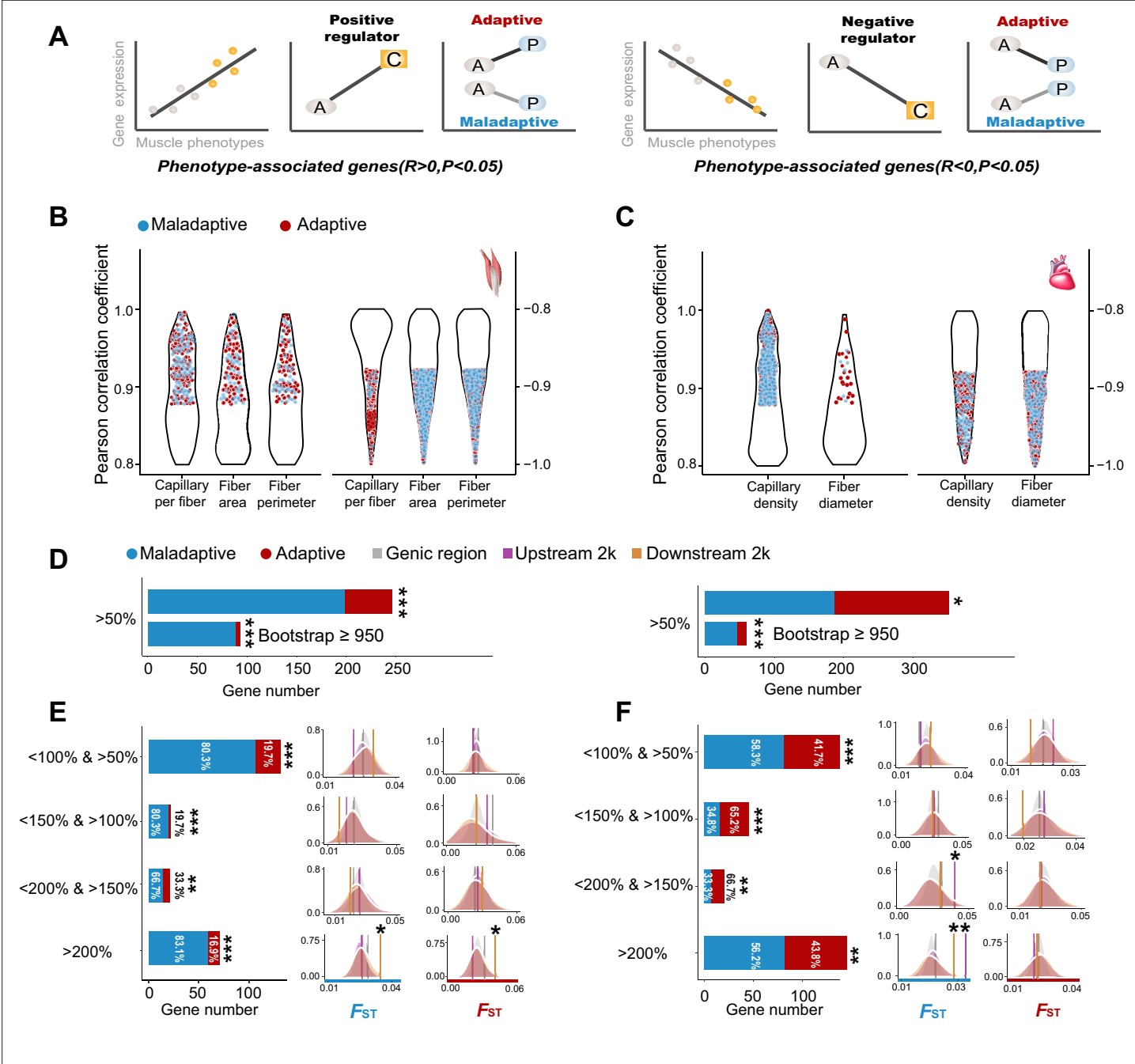

**Figure 3.** Genes with expression levels associated with muscle phenotypes and adaptive and maladaptive plasticity. (**A**) Gene expression profiles were correlated with muscle phenotypes. The direction of gene expression plasticity at the plastic stage matched the expectation for positive regulator (increased expression) and negative regulators (decreased expression) in the colonized tree sparrows was considered to indicate adaptive plasticity. Conversely, those that opposed this expectation were considered to be maladaptive, i.e., the direction of gene expression plasticity at the plastic stage showed decreased expression for the positive regulators and increased expression for negative regulators. A, ancestral stage; P, plastic stage; C, colonized stage. (**B**) and (**C**) Gene expression profiles were correlated with three flight muscle phenotypes (B, capillary per fiber, fiber area, and perimeter) and two cardiac muscle phenotype (C, capillary density, and fiber diameter). Red dots show genes with adaptive gene expression plasticity and blue dots show genes with maladaptive gene expression plasticity. (**D**) Frequencies of genes with adaptive and maladaptive plasticity (>50%) and their subsets that acquire strong support in the parametric bootstrap analyses (≥950/1000). (**E**) and (**F**) Left, candidate genes are classified into four categories with different magnitudes of adaptive or maladaptive plasticity. More genes show maladaptive plasticity than those that show adaptive plasticity. Two-tailed binomial test, **p<0.01; ***p<0.001. Right, the empirical $F_{ST}$ values significantly increased in the 2 k upstream and/or downstream regions of the genes having adaptive/maladaptive plasticity <200% and >150% (cardiac muscle) and >200% (flight and cardiac muscles). (**E**) Flight

*Figure 3 continued on next page*

*Figure 3 continued*

muscle, (**F**) Cardiac muscle. Vertical lines, the empirical $F_{ST}$ values; shades, permuted $F_{ST}$ distributions generated from 100 times sampling. *$p<0.05$; **$p<0.01$.

The online version of this article includes the following source data and figure supplement(s) for figure 3:

**Source data 1.** Raw data for *Figure 3B–F*.

**Figure supplement 1.** Pearson correlation between gene expression and muscle phenotypic values of the ancestral and colonized individuals.

**Figure supplement 2.** Frequencies of genes with adaptive and maladaptive plasticity in the flight and cardiac muscles.

colonized tree sparrows tends to increase in the 2 kb upstream and downstream regions of these genes ($p<0.05$, *Figure 3E–F*). These results suggest that selection on regulatory adaptation (i.e. genetic divergence) may only be expected if large strength of selection is needed to bring initial plasticity toward adaptive optimum of the novel environment.

## Discussion

An understanding of phenotypic plasticity and its consequence for adaptive evolution is important in evolutionary biology but remains challenging (*Ghalambor et al., 2007*; *Ghalambor et al., 2015*; *Campbell-Staton et al., 2021*; *Kenkel and Matz, 2016*; *Rivera et al., 2021*). Using an integrative approach of field-based studies, experimentally implemented acclimation experiments, multiomic and muscle phenotypic data, our study demonstrates the importance of gene expression plasticity in facilitating initial survival and population persistence at an early stage of colonization. In particular, our results provide novel insight into an issue of active debate within evolutionary biology (i.e. reinforcement vs. reversion plasticity) by showing that the evolutionary consequence of plasticity depends on the adaptive plasticity (reinforcement) or maladaptive plasticity (reversion). We also demonstrate that natural selection drives genetic divergence when the magnitude of reinforcement and/or reversion plasticity becomes intense, providing novel insights into the mechanism of plasticity first and genes as followers *Schwander and Leimar, 2011*.

Our gene expression and sequence analyses provide several lines of evidence in support of gene expression plasticity contribution to the high-elevation colonization of tree sparrows. First, we show that co-expressed genes and muscle phenotype-associated genes change either the intensity or the direction of their expression profiles, and that gene expressions at the plastic stage are more often reversed than reinforced at the colonization stage. Second, by combining gene expression and muscle phenotype, we show that selection operates intensely on maladaptive plasticity during the initial stage of responding novel environment (*Ghalambor et al., 2007*; *Ghalambor et al., 2015*; *Ho and Zhang, 2018*; *Campbell-Staton et al., 2021*). Third, the genes with intensified levels of reinforcement and reversion expression plasticity show large genetic divergence ($F_{ST}$) in their upstream and downstream regions between the ancestral and colonized populations, suggesting that selection has driven genetic changes in the noncoding regions after the tree sparrows had colonized the novel environment.

These observations thus support that plasticity serves as a stepping stone in a successful colonization of a new environment and that genetic changes that evolve at a late stage will modify the plastic change to an optimal phenotype in the new environment (*Corl et al., 2018*; *Schwander and Leimar, 2011*). It is interesting to think about why phenotypic change at the plastic stage sometimes differs from those at the colonized stage. A likely reason is that initially plastic change allows organisms to survive upon a sudden environmental shift but the fitness is much reduced compared to that after a long-term colonization to the novel environment (*Fischer et al., 2016*; *Huang and Agrawal, 2016*; *Leonard and Lancaster, 2020*; *Kuo et al., 2023*). Thus, the overall physiological states of the organisms right after an environmental shift may result in low fitness. To coordinate the phenotype values to close the organismal fitness, a genetic change may be required to modify the plastic change of phenotype to approach the optimum to the novel environment (*Storz, 2021*). Thus, the consequence of genetic change on plasticity is dependent on the cost of plastic change (*Ho and Zhang, 2018*; *Corl et al., 2018*).

A limitation of our study is that we could only consider hypoxic conditions in the acclimation experiment, while also other factors associated with high-elevation environments, e.g., low temperature

and UV radiation, may be of importance. Although we assume that hypoxia is the dominant selective force in the Qinghai-Tibetan Plateau (see also *Sun et al., 2016*; *Qu et al., 2020*), the use of a single environmental stimulus makes our conclusions more conservative. Also, it is reasonable to assume that plastic change in organisms is time-dependent and that our results cannot catch gene expression occurring at the later stage of plastic response (i.e. after one month). Since we could only run the acclimation experiment for one month it is likely we have not identified all genes related to hypoxia tolerance. Despite this, we could observe a clear pattern of reinforcement and reversion plasticity and the associated genetic changes. Our sample sizes are rather small (n=12 for the flight muscle and n=9 for the cardiac muscle), mostly because of the logistical challenge of keeping tree sparrows in the necessary common garden experiments. In order to compensate for this, we utilized comparative transcriptomics to generate the four independent datasets (a group of co-expressed genes and a group of phenotype-associated genes for the flight and cardiac muscles, respectively). As the analyses of these four independent datasets show similar results, we believe the conclusions drawn from our study are robust.

Despite some limitations, our study demonstrates an easy implemented framework for quantifying phenotype plasticity and how to relate this to genetic change. Short-term acclimation response of lowland populations has been found to be different from the populations adapted to the high-elevation environments. For example, people newly exposed to high-elevation environment increase their red blood cells as compared to people at sea level, which is likely mediated by *HIF* and *EPO* genes in the hypoxia response cascade (*Erzurum et al., 2007*; *Peng et al., 2017*). This physiological change, however, is less activated in native people at high-elevation when compared to people at sea level, demonstrating an example of reversion of phenotypic plasticity (*Storz, 2021*). Together with previous studies, our work shows that short-term acclimation response and long-term adaptive evolution in hypoxia response may affect different evolutionary pathways. Our work provides an understanding of how plasticity facilitates species to invade new habitats and survive environmental change. Such an understanding is important, in particular when numerous species are being introduced to new regions of the globe through human intervention and spread as invasive species (*Visser, 2008*).

## Materials and methods
### Sampling
We generated transcriptomic data from hypoxia-acclimated lowland tree sparrows by exposing five lowland tree sparrows to hypoxic conditions for 30 days using a hypoxic chamber (*Qu et al., 2020*). The oxygen content in the chamber was set to 14% of the oxygen content (simulating the oxygen concentration at 3200 m a.s.l., 70% of ~20%, the latter of which is the content of oxygen content at sea level), which is similar to the condition where we collected the highland tree sparrows (Qinghai Lake, 3200 m a.s.l.). We collected flight muscle from the four individuals and cardiac muscle from three individuals. In addition, we extended the analyses of the transcriptomic and muscle histological data from tree sparrows that were collected from lowland (Beijing and Hebei, 100 m a.s.l.) and highland (Qinghai Lake, 3200 m a.s.l., *Figure 1*) collected in previous study (*Qu et al., 2020*). In total, we used twelve samples for flight muscle and nine samples for cardiac muscle (*Supplementary file 1*). The ancestral lowland and hypoxia-exposed lowland tree sparrows were collected from the same locality (Beijing) and the same season (pre-breeding). Only adult birds with similar body weight (approximately 18 g) were used.

### Transcriptome sequencing
RNA libraries were constructed for the flight and cardiac muscles and sequenced on an Illumina HiSeq4000 platform. After filtering low-quality, adapter-contaminated, and N-rich reads (>10%), a total of 134, 128, and 120 million reads of transcriptional data were generated for the lowland, hypoxia-exposed lowland tree sparrows and highland tree sparrows, respectively (*Supplementary file 1*). We mapped cleaned reads against the tree sparrow genome using STAR (*Dobin et al., 2013*). We compared log-transformed transcript per million (TPM) across all genes and determined the most conserved genes (i.e. coefficient of variance $\leq 0.3$ and average TPM $\geq 1$ for each sample) for the flight and cardiac muscles, respectively (*Hao et al., 2023*). We then compared the median expression levels of these conserved genes and found no difference among the lowland, hypoxia-exposed lowland,

and highland tree sparrows (Wilcoxon signed-rank test, p<0.05, *Figure 2—figure supplement 3*), suggesting that batch effect had little influence on the transcriptomic data. We then used TPM values to calculate gene expression level and intensity using RSEM (*Li and Dewey, 2011*).

## WGCNA analyses

We used WGCNA v. 1.41–1 (*Langfelder and Horvath, 2008*) to identify regulatory architecture for the flight muscle and cardiac muscle transcriptomes. Specifically, we used a principal component analysis (PCA) to summarize modules of genes expression with blockwiseModules function, and then used module eigengene values of the first principal component (PC) to test the correlation between module expression and stages. We used Pearson linear correlation (cor function) to identify genes with expression reinforcement and nonlinear regression (polynomial regression model: Model = lmy~poly(x, 5, raw = TRUE), data = data) to identify genes with expression reversion. Regulatory modules were identified as branches of the resulting cluster tree via the dynamic tree-cutting method and highly correlated modules (p<0.1) were merged. We used 'GS>first quartile of GS' and 'p<0.05' to identify important genes within modules (hub genes).

## Identify gene regulation associated with muscle phenotype

To identify the genes with expression levels that correlated to evolved differences in the muscle phenotypes in the tree sparrows, we used the lowland and highland individuals from *Qu et al., 2020* for which RNAseq and muscle phenotypic data were available. We correlated the levels of gene expression with muscle phenotypes in the lowland and highland individuals. We used three phenotypes for the flight muscle (capillary per fiber, fiber area, and perimeter) and two for the cardiac muscle (capillary density and fiber diameter). We used Pearson correlation to test the association between each of the phenotypic traits and the expression profile of each gene (TPM). A threshold of p<0.05 was used to detect muscle-associated genes. We then followed *Campbell-Staton et al., 2021* to categorize genes as positive (positively correlated with muscle phenotypes) or negative regulators (negatively correlated with muscle phenotypes).

Subsequently, we compared the direction of gene expression plasticity at the plastic stage with those of the positive and negative regulators. Briefly, the direction of gene expression plasticity at the plastic stage matched the expectation for the positive regulator (increased expression) or negative regulator (decreased expression) in the colonized tree sparrows was considered to indicate adaptive plasticity. Conversely, those that opposed this expectation were considered to be maladaptive, i.e., the direction of gene expression plasticity at the plastic stage showed decreased expression for the positive regulators and increased expression for negative regulators (*Figure 3A*).

## Reinforcement and reversion plasticity analyses

To investigate the magnitude of reinforcement and reversion plasticity, we followed the method described by *Ho and Zhang, 2018*. Specifically, expression levels of each gene in the three stages were treated as E_lowland (expression level at the ancestral stage), E_hypoxia (expression level at the plastic stage), and E_highland (expression level at the colonized stage). We used a threshold of 50% of the ancestral gene expression level, i.e., 50% * E_lowland, to detect an excess of gene expression change. We identified genes with an excess of plastic change if it satisfied the condition of |(E_hypoxia-E_lowland)|>50% * E_lowland. Likewise, we identified genes with an excess of evolutionary change if it met the condition of |(E_highland-E_hypoxia)|>50%* E_lowland. If the genes showed the same direction in the expression change at the plastic and colonized stages, we regarded these genes as reinforcement expression. If the genes showed opposite directions in the expression change at the plastic and colonized stages, we regarded these genes as reversion expression. We categorized these genes into reinforcement and reversion groups and then tested if the proportions of the genes in the two groups were significantly different from the expected proportions using two-tailed binomial tests.

To explore whether an excess of genes with reversion plasticity was subject to random sampling errors (*Mallard et al., 2018*), we used a parametric bootstrap method as described in *Ho and Zhang, 2019*. Specifically, we simulated the mean expression level of a gene at each stage following a Gaussian distribution within the mean equal to the observed mean expression of this gene at this stage and the standard deviation equal to the estimated standard error. We then draw a random variable from the above Gaussian distribution to represent an observation of the mean expression level of this gene

at this stage. We drew random variables representing the mean expression level of genes at each of the three stages (i.e. ancestral, plastic, and colonized stages), and then computed gene expression changes at the plastic stage and colonized stage and determined the reinforcement or reversion plasticity. This process is repeated 1000 times for each gene. If this gene exhibited reinforcement or reversion plasticity at least 950 repeats (p<0.05), we considered this gene showing reinforcement or reversion plasticity. The Gaussian distribution and random sampling were generated using the rnorm function of R.

To test the impact of the choice of thresholds on the results, we analyzed the data using a range of thresholds, i.e., <100% and >50%, <150% and >100%, <200% and >150%, and >200%. This setting aimed to obtain independent genesets for subsequent genetic divergence analyses, because a geneset selected by relax threshold (i.e. 50%) also included genesets obtained by the stringent threshold (i.e. 100%, 150%, and 200%). In addition, we also implemented other categorization scheme to check the robustness of the results. Specifically, we used three different bin settings (i.e. 20%, 40%, and 60% bin settings along the spectrum of the reinforcement/reversion plasticity), to group genes according to their magnitude of plasticity. For each category in the three-bin settings, we compared whether the proportion of the genes with reversion plasticity differed from that with reinforcement plasticity using two-tailed binomial tests.

## Comparing genetic divergence for genes with reinforcement and reversion expression plasticity

To calculate the genetic divergence between the lowland ancestral and highland colonized tree sparrows, we used 12 lowland and 11 highland individuals for which re-sequencing data were available (*Qu et al., 2020*). After mapping the raw reads to the tree sparrow genome using BWA v0.7.17 (*Li and Durbin, 2009*), we obtained mean sequence coverage of 17x for each individual (*Supplementary file 2*). We called variants with GATK v 3.7 (*McKenna et al., 2010*) and Samtools v1.2 (http://www.htskib.org/) and filtered single nucleotide polymorphism (SNP) using VCF tools and GATK with minimum coverage = 138, root mean square mapping quality ≥20, distance of adjacent SNPs ≥5, distance to a gap ≥5 bp, and read quality value ≥30.

We calculated average $F_{ST}$ using Vcftools (*Danecek et al., 2011*) with SNPs included in the genic region, 2 kb upstream and downstream regions of the genes within each of the four categories, i.e., <100% and >50%, <150% and>100%, <200% and>150%, and >200%. The upstream and downstream regions were considered because gene regulation may refer to genetic variation in the nearby genic regions. We permuted $F_{ST}$ distributions for the genes within each category by random sampling the same number of SNPs, allowing for a fluctuation of 5% of total SNPs. We compared the empirical $F_{ST}$ values to permuted $F_{ST}$ distributions generated from 100 samplings and considered a threshold of p<0.05 to be statistically significant (empirical $F_{ST}$ >95% percentile of permuted $F_{ST}$ distribution). We calculated empirical $F_{ST}$ values and permutated $F_{ST}$ distributions for the genic region, 2 kb upstream and downstream regions separately. To avoid the sampling bias (i.e. sampling SNPs from neutral regions), our SNP sampling was constrained to the genic region, 2 k upstream and downstream regions of all genes (i.e. 16,925 genes). All analyses were conducted with the R statistical software package (*R Development Core Team, 2017*).

## Acknowledgements

We acknowledge Ying Xiong for logistic work in the hypoxia-exposed experiment. This research was funded by the Third Xinjiang Scientific Expedition and Research (XIKK) (2022xjkk0205), and the National Natural Science Foundation of China (32020103005 and U23A20162).

# Additional information

## Funding

| Funder | Grant reference number | Author |
|--------|------------------------|--------|
| Ministry of Science and Technology of the People's Republic of China | Third Xinjiang Scientific Expedition and Research 2022xjkk0205 | Yanhua Qu |
| Ministry of Science and Technology of the People's Republic of China | Second Tibetan Plateau Scientific Expedition and Research Yanhua Qu 2019QZKK0501 | Yanhua Qu |
| National Natural Science Foundation of China | National Natural Science Foundation of China Yanhua Qu NSFC32020103005 | Yanhua Qu |

The funders had no role in study design, data collection and interpretation, or the decision to submit the work for publication.

## Author contributions

Huishang She, Data curation, Formal analysis, Investigation, Visualization, Methodology, Writing - original draft, Writing - review and editing; Yan Hao, Formal analysis, Methodology; Gang Song, Xu Luo, Weiwei Zhai, Data curation, Investigation; Fumin Lei, Resources, Data curation; Yanhua Qu, Conceptualization, Resources, Funding acquisition, Investigation, Writing - original draft, Writing - review and editing

## Author ORCIDs

Huishang She (ID) http://orcid.org/0000-0001-6259-7904
Yanhua Qu (ID) http://orcid.org/0000-0002-4590-7787

Reviewer #1 (Public Review): https://doi.org/10.7554/eLife.86687.3.sa1
Reviewer #2 (Public Review): https://doi.org/10.7554/eLife.86687.3.sa2
Author Response https://doi.org/10.7554/eLife.86687.3.sa3

---

# Additional files

## Supplementary files

• Supplementary file 1. Sampling of transcriptomic data used in this study. All birds collected were adults and at pre-breeding season. Sex was unknown.

• Supplementary file 2. Statistics of reads mapping and coverage of 11 highland and 12 lowland tree sparrows (*Qu et al., 2020*). All birds collected were adults and sex is unknown.

• MDAR checklist

## Data availability

DNA sequencing reads used in this study have been deposited in Short Read Archive under the project number PRJNA417520. The custom code is available at https://github.com/shelfey/tree-sparrow-acclimation/blob/main/TScode.R (copy archived at *shelfey, 2024*).

The following dataset was generated:

| Author(s) | Year | Dataset title | Dataset URL | Database and Identifier |
|-----------|------|---------------|-------------|-------------------------|
| She H, Hao Y, Song G, Luo X, Lei F, Zhai W, Qu Y | 2017 | Genome sequencing and assembly of three snow finches and one tree sparrows | https://www.ncbi.nlm.nih.gov/bioproject/PRJNA417520 | NCBI BioProject, PRJNA417520 |

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
