## [Editor Report · eLife assessment]

This study provides **useful** information on the evolution of gene expression levels and plasticity in tissues impacted by hypoxia during colonization of a high-altitude environment. Unfortunately, the evidence for the conclusions is **incomplete** because of the low sample size available.

---

## [Referee Report · Reviewer #1 (Public Review)]

She et al studied the evolution of gene expression reaction norms when individuals colonise a new environment that exposes them to physiologically challenging conditions. Their objective was to test the "plasticity first" hypothesis, which suggest that traits that are already plastic (their value changes when facing a new environment compared to the original environment) facilitates the colonisation of novel environments, which, if true, would be predicted to result in the evolution of gene expression values that are similar in the population that colonised the new environment and evolved under these particular selection pressures. To test this prediction, they studied gene expression in cardiac and muscle tissues in individuals originating from three conditions: lowland individuals in their natural environment (ancestral state), lowland individuals exposed to hypoxia (the plastic response state), and a highland population facing hypoxia for several generations (the coloniser state). They classified gene expression patterns as maladaptive or adaptive in lowland individuals responding to short term hypoxia by classifying gene expression patterns using genes that differed between the ancestral state (lowland) and colonised state (highland). Genes expressed in the same direction in lowland individuals facing hypoxia (the plastic state) as what is found in the colonised state are defined as adaptative, while genes with the opposite expression pattern were labelled as maladaptive, using the assumption that the colonised state must represent the result of natural selection. Furthermore, genes could be classified as representing reversion plasticity when the expression pattern differed between the plasticity and colonised states and as reinforcement when they were in the same direction (for example more expressed in the plastic state and the colonised state than in the ancestral state). They found that more genes had a plastic expression pattern that was labelled as maladaptive than adaptive. Therefore, some of the genes have an expression pattern in accordance with what would be predicted based on the plasticity-first hypothesis, while others do not.

As pointed out by the authors themselves, the fact that temperature was not included as a variable, which would make the experimental design much more complex, misses the opportunity to more accurately reflect the environmental conditions that the colonizer individuals face at high altitude. Also pointed out by the authors, the acclimation experiment in hypoxia lasted 4 weeks. It is possible that longer term effects would be identifiable in gene expression in the lowland individuals facing hypoxia on a longer time scale. Furthermore, a sample size of 3 or 4 individuals per group depending on the tissue for wild individuals may miss some of the natural variation present in these populations. Stating that they have a n=7 for the plastic stage and n = 14 for the ancestral and colonized stages refers to the total number of tissue samples and not the number of individuals, according to supplementary table 1.

Impact of the work:

There has been work showing that populations adapted to high altitude environments show changes in their hypoxia response that differs from the short-term acclimation response of lowland population of the same species. For example, in humans, see Erzurum et al. 2007 and Peng et al. 2017, where they show that the hypoxia response cascade, which starts with the gene HIF (Hypoxia-Inducible Factor) and includes the EPO gene, which codes for erythropoietin, which in turns activates the production of red blood cell, is LESS activated in high altitude individuals compared to the activation level in lowland individuals (which gives it its name). The present work adds to this body of knowledge showing that the short-term response to hypoxia and the long term one can affect different pathways and that acclimation/plasticity does not always predict what physiological traits will evolve in populations that colonize these environments over many generations and additional selection pressure (UV exposure, temperature, nutriment availability).

Altogether, this work provides new information on the evolution of reaction norms of genes associated with the physiological response to one of the main environmental variables that affects almost all animals, oxygen availability. It also provides an interesting model system to study this type of question further in a natural population of homeotherms.

Erzurum, S. C., S. Ghosh, A. J. Janocha, W. Xu, S. Bauer, N. S. Bryan, J. Tejero et al. "Higher blood flow and circulating NO products offset high-altitude hypoxia among Tibetans." Proceedings of the National Academy of Sciences 104, no. 45 (2007): 17593-17598.

Peng, Y., C. Cui, Y. He, Ouzhuluobu, H. Zhang, D. Yang, Q. Zhang, Bianbazhuoma, L. Yang, Y. He, et al. 2017. Down-regulation of EPAS1 transcription and genetic adaptation of Tibetans to high-altitude hypoxia. Molecular biology and evolution 34:818-830.

---

## [Referee Report · Reviewer #2 (Public Review)]

This is a well-written paper using gene expression in tree sparrow as model traits to distinguish between genetic effects that either reinforce or reverse initial plastic response to environmental changes. Tree sparrow tissues (cardiac and flight muscle) collected in lowland and highland populations subject to hypoxia treatment were profiled for gene expression and compared in (1) highland birds; (2) lowland birds under normal conditions to test for differences in directions of changes between initial plastic response and subsequent colonized response.

The authors clarified several points and made revisions according to my comments. It is good to know that the highland and lowland samples were collected and processed at the same time and the previous publication reported part of the data. My concerns regarding the conclusions about reversal versus reinforcement remain even after the additional analyses. Further studies are needed to confirm these results.

---

## [Author Response]

The following is the authors’ response to the original reviews.

**Reviewer #1 (Public Review):**
She et al studied the evolution of gene expression reaction norms when individuals colonise a new environment that exposes them to physiologically challenging conditions. Their objective was to test the "plasticity first" hypothesis, which suggest that traits that are already plastic (their value changes when facing a new environment compared to the original environment) facilitates the colonisation of novel environments, which, if true, would be predicted to result in the evolution of gene expression values that are similar in the population that colonised the new environment and evolved under these particular selection pressures. To test this prediction, they studied gene expression in cardiac and muscle tissues in individuals originating from three conditions: lowland individuals in their natural environment (ancestral state), lowland individuals exposed to hypoxia (the plastic response state), and a highland population facing hypoxia for several generations (the coloniser state). They classified gene expression patterns as maladaptive or adaptive in lowland individuals responding to short term hypoxia by classifying gene expression patterns using genes that differed between the ancestral state (lowland) and colonised state (highland). Genes expressed in the same direction in lowland individuals facing hypoxia (the plastic state) as what is found in the colonised state are defined as adaptative, while genes with the opposite expression pattern were labelled as maladaptive, using the assumption that the colonised state must represent the result of natural selection. Furthermore, genes could be classified as representing reversion plasticity when the expression pattern differed between the plasticity and colonised states and as reinforcement when they were in the same direction (for example more expressed in the plastic state and the colonised state than in the ancestral state). They found that more genes had a plastic expression pattern that was labelled as maladaptive than adaptive. Therefore, some of the genes have an expression pattern in accordance with what would be predicted based on the plasticity-first hypothesis, while others do not.

Thank you for a precise summary of our work. We appreciate the very encouraging comments recognizing the value of our work. We have addressed concerns from the reviewer in greater detail below.

Q1. As pointed out by the authors themselves, the fact that temperature was not included as a variable, which would make the experimental design much more complex, misses the opportunity to more accurately reflect the environmental conditions that the colonizer individuals face at high altitude. Also pointed out by the authors, the acclimation experiment in hypoxia lasted 4 weeks. It is possible that longer term effects would be identifiable in gene expression in the lowland individuals facing hypoxia on a longer time scale. Furthermore, a sample size of 3 or 4 individuals per group depending on the tissue for wild individuals may miss some of the natural variation present in these populations. Stating that they have a n=7 for the plastic stage and n = 14 for the ancestral and colonized stages refers to the total number of tissue samples and not the number of individuals, according to supplementary table 1.

We shared the same concerns as the reviewer. This is partly because it is quite challenging to bring wild birds into captivity to conduct the hypoxia acclimation experiments. We had to work hard to perform acclimation experiments by taking lowland sparrows in a hypoxic condition for a month. We indeed have recognized the similar set of limitations as the review pointed out and have discussed the limitations in the study, i.e., considering hypoxic condition alone, short time acclimation period, etc. Regarding sample sizes, we have collected cardiac muscle from nine individuals (three individuals for each stage) and flight muscle from 12 individuals (four individuals for each stage). We have clarified this in Supplementary Table 1.

Q2. Finally, I could not find a statement indicating that the lowland individuals placed in hypoxia (plastic stage) were from the same population as the lowland individuals for which transcriptomic data was already available, used as the "ancestral state" group (which themselves seem to come from 3 populations Qinghuangdao, Beijing, and Tianjin, according to supplementary table 2) nor if they were sampled in the same time of year (pre reproduction, during breeding, after, or if they were juveniles, proportion of males or females, etc). These two aspects could affect both gene expression (through neutral or adaptive genetic variation among lowland populations that can affect gene expression, or environmental effects other than hypoxia that differ in these populations' environments or because of their sexes or age). This could potentially also affect the FST analysis done by the authors, which they use to claim that strong selective pressure acted on the expression level of some of the genes in the colonised group.

The reviewer asked how individual tree sparrows used in the transcriptomic analyses were collected. The individuals used for the hypoxia acclimation experiment and represented the ancestral lowland population were collected from the same locality (Beijing) and at the same season (i.e., pre-breeding) of the year. They are all adults and weight approximately 18g. We have clarified this in the Supplementary Table S1 and Methods. We did not distinguish males from females (both sexes look similar) under the assumption that both sexes respond similarly to hypoxia acclimation in their cardiac and flight muscle gene expression.

The Supplementary Table 2 lists the individuals that were used for sequence analyses. These individuals were only used for sequence comparisons but not for the transcriptomic analyses. The population genetic structure analyzed in a previously published study showed that there is no clear genetic divergence within the lowland population (i.e., individuals collected from Beijing, Tianjing and Qinhuangdao) or the highland population (i.e., Gangcha and Qinghai Lake). In addition, there was no clear genetic divergence between the highland and lowland populations (Qu et al. 2020).

Q4. Impact of the workThere has been work showing that populations adapted to high altitude environments show changes in their hypoxia response that differs from the short-term acclimation response of lowland population of the same species. For example, in humans, see Erzurum et al. 2007 and Peng et al. 2017, where they show that the hypoxia response cascade, which starts with the gene HIF (Hypoxia-Inducible Factor) and includes the EPO gene, which codes for erythropoietin, which in turns activates the production of red blood cell, is LESS activated in high altitude individuals compared to the activation level in lowland individuals (which gives it its name). The present work adds to this body of knowledge showing that the short-term response to hypoxia and the long term one can affect different pathways and that acclimation/plasticity does not always predict what physiological traits will evolve in populations that colonize these environments over many generations and additional selection pressure (UV exposure, temperature, nutrient availability). Altogether, this work provides new information on the evolution of reaction norms of genes associated with the physiological response to one of the main environmental variables that affects almost all animals, oxygen availability. It also provides an interesting model system to study this type of question further in a natural population of homeotherms.Erzurum, S. C., S. Ghosh, A. J. Janocha, W. Xu, S. Bauer, N. S. Bryan, J. Tejero et al. "Higher blood flow and circulating NO products offset high-altitude hypoxia among Tibetans." Proceedings of the National Academy of Sciences 104, no. 45 (2007): 17593-17598.Peng, Y., C. Cui, Y. He, Ouzhuluobu, H. Zhang, D. Yang, Q. Zhang, Bianbazhuoma, L. Yang, Y. He, et al. 2017. Down-regulation of EPAS1 transcription and genetic adaptation of Tibetans to high-altitude hypoxia. Molecular biology and evolution 34:818-830.

Thank you for highlighting the potential novelty of our work in light of the big field. We found it very interesting to discuss our results (from a bird species) together with similar findings from humans. In the revised version of manuscript, we have discussed short-term acclimation response and long-term adaptive evolution to a high-elevation environment, as well as how our work provides understanding of the relative roles of short-term plasticity and long-term adaptation. We appreciate the two important work pointed out by the reviewer and we have also cited them in the revised version of manuscript.

**Reviewer #2 (Public Review):**
This is a well-written paper using gene expression in tree sparrow as model traits to distinguish between genetic effects that either reinforce or reverse initial plastic response to environmental changes. Tree sparrow tissues (cardiac and flight muscle) collected in lowland populations subject to hypoxia treatment were profiled for gene expression and compared with previously collected data in (1) highland birds; (2) lowland birds under normal condition to test for differences in directions of changes between initial plastic response and subsequent colonized response. The question is an important and interesting one but I have several major concerns on experimental design and interpretations.

Thank you for a precise summary of our work and constructive comments to improve this study. We have addressed your concerns in greater detail below.

Q1. The datasets consist of two sources of data. The hypoxia treated birds collected from the current study and highland and lowland birds in their respective native environment from a previous study. This creates a complete confounding between the hypoxia treatment and experimental batches that it is impossible to draw any conclusions. The sample size is relatively small. Basically correlation among tens of thousands of genes was computed based on merely 12 or 9 samples.

We appreciate the critical comments from the reviewer. The reviewer raised the concerns about the batch effect from birds collected from the previous study and this study. There is an important detail we didn’t describe in the previous version. All tissues from hypoxia acclimated birds and highland and lowland birds have been collected at the same time (i.e., Qu et al. 2020). RNA library construction and sequencing of these samples were also conducted at the same time, although only the transcriptomic data of lowland and highland tree sparrows were included in Qu et al. (2020). The data from acclimated birds have not been published before.

In the revised version of manuscript, we also compared log-transformed transcript per million (TPM) across all genes and determined the most conserved genes (i.e., coefficient of variance ≤ 0.3 and average TPM ≥ 1 for each sample) for the flight and cardiac muscles, respectively (Hao et al. 2023). We compared the median expression levels of these conserved genes and found no difference among the lowland, hypoxia-exposed lowland, and highland tree sparrows (Wilcoxon signed-rank test, P<0.05). As these results suggested little batch effect on the transcriptomic data, we used TPM values to calculate gene expression level and intensity. This methodological detail has been further clarified in the Methods and we also provided a new supplementary Figure (Figure S5) to show the comparative results.

The reviewer also raised the issue of sample size. We certainly would have liked to have more individuals in the study, but this was not possible due to the logistical problem of keeping wild bird in a common garden experiment for a long time. We have acknowledged this in the manuscript. In order to mitigate this we have tested the hypothesis of plasticity following by genetic change using two different tissues (cardiac and flight muscles) and two different datasets (co-expressed gene-set and muscle-associated gene-set). As all these analyses show similar results, they indicate that the main conclusion drawn from this study is robust.

Q2. Genes are classified into two classes (reversion and reinforcement) based on arbitrarily chosen thresholds. More "reversion" genes are found and this was taken as evidence reversal is more prominent. However, a trivial explanation is that genes must be expressed within a certain range and those plastic changes simply have more space to reverse direction rather than having any biological reason to do so.

Thank you for the critical comments. There are two questions raised we should like to address them separately. The first concern centered on the issue of arbitrarily chosen thresholds. In our manuscript, we used a range of thresholds, i.e., 50%, 100%, 150% and 200% of change in the gene expression levels of the ancestral lowland tree sparrow to detect genes with reinforcement and reversion plasticity. By this design we wanted to explore the magnitudes of gene expression plasticity (i.e., Ho & Zhang 2018), and whether strength of selection (i.e., genetic variation) changes with the magnitude of gene expression plasticity (i.e., Campbell-Staton et al. 2021).

As the reviewer pointed out, we have now realized that this threshold selection is arbitrarily. We have thus implemented two other categorization schemes to test the robustness of the observation of unequal proportions of genes with reinforcement and reversion plasticity. Specifically, we used a parametric bootstrap procedure as described in Ho & Zhang (2019), which aimed to identify genes resulting from genuine differences rather than random sampling errors. Bootstrap results suggested that genes exhibiting reversing plasticity significantly outnumber those exhibiting reversing plasticity, suggesting that our inference of an excess of genes with reversion plasticity is robust to random sampling errors. We have added these analyses to the revised version of manuscript, and provided results in the Figure 2d and Figure 3d.

In addition, we adapted a bin scheme (i.e., 20%, 40% and 60% bin settings along the spectrum of the reinforcement/reversion plasticity). These analyses based on different categorization schemes revealed similar results, and suggested that our inference of an excess of genes with reversion plasticity is robust. We have provided these results in the Supplementary Figure S2 and S4.

The second issue that the reviewer raised is that the plastic changes simply have more space to reverse direction rather than having any biological reason to do so. While a causal reason why there are more genes with expression levels being reversed than those with expression levels being reinforced at the late stages is still contentious, increasingly many studies show that genes expression plasticity at the early stage may be functionally maladapted to novel environment that the species have recently colonized (i.e., lizard, Campbell-Staton et al. 2021; *Escherichia coli*, yeast, guppies, chickens and babblers, Ho and Zhang 2018; Ho et al. 2020; Kuo et al. 2023). Our comparisons based on the two genesets that are associated with muscle phenotypes corroborated with these previous studies and showed that initial gene expression plasticity may be nonadaptive to the novel environments (i.e., Ghalambor et al.2015; Ho & Zhang 2018; Ho et al. 2020; Kuo et al. 2023; Campbell-Staton et al. 2021).

Q3. The correlation between plastic change and evolved divergence is an artifact due to the definitions of adaptive versus maladaptive changes. For example, the definition of adaptive changes requires that plastic change and evolved divergence are in the same direction (Figure 3a), so the positive correlation was a result of this selection (Figure 3d).

The reviewer raised an issue that the correlation between plastic change and evolved divergence is an artifact because of the definition of adaptive versus maladaptive changes, for example, Figure 3d. We agree with the reviewer that the correlation analysis is circular because the definition of adaptive and maladaptive plasticity depends on the direction of plastic change matched or opposed that of the colonized tree sparrows. We have thus removed previous Figure 3d-e and related texts from the revised version of manuscript. Meanwhile, we have changed Figure 3a to further clarify the schematic framework.

**Reviewer #1 (Recommendations For The Authors):**
Q1. Here are private recommendations that I think could help improve the manuscript.West-Eberhard was a pioneer back in 2003 in explicating the hypothesis of "plasticity first". I think it is important to cite their main work in the first paragraph of introduction and to use the term "plasticity-first", which is widely known among evolutionary biologists studying phenotypic plasticity, instead of "plasticity followed by genetic change", since the three papers cited in paragraph 1 call it « plasticity first ».West-Eberhard, M.J. (2003) Developmental Plasticity and Evolution, Oxford University Press.

Thank you for suggesting West-Eberhard (2003) and we have cited this important work. We have also changed “plasticity followed by genetic change” to “plasticity first”.

Q2. Introduction. Line 5, Change for « On the one hand, if plasticity changes ... »

We have modified as suggested.

Q3. Line 52, Change for « ...same direction as adaptive evolution does ...»

We have modified as suggested.

Q4. Line 66,When presenting papers that address the plasticity and evolution of gene expression in response to environmental variables, paper by Morris et al is another example that could be useful to include (but this is only a suggestion in case the authors missed it).

Thank you for suggesting this nice work. We have cited Morris et al. (2014).

Q5. Line 94, Change for "We acclimated"

We have modified as suggested.

Q6. In Figure 3, the figure in panel A and B is labelled "normaxia", but I think that "normoxia" is usually the term used.

Thank you for spot the typo. We have modified Figure 3a and we no longer used the term “normaxia”.

Material and methodsIt would be important to merge supplementary table 1 and 2 and only present the individuals that were used with their respective cardiac and muscle libraries (if they come from the same individual?). Also, the origin of the individuals used in the hypoxia experiment should be explained at the beginning of the methods section and explicated in the supplementary table. Information on sex or stage of development (juvenile? Adult? Male? female?) and time of year (in breeding stage? Pre-migration (if any), etc) would allow the reader to see that individuals from lowland differed only in their exposure to hypoxia or not, or if other variables may affect gene expression patterns. Similarly, if all individuals form the highland are males and the lowland hypoxia exposed individuals are females (or juveniles versus breeders, or different time of year, etc) this should be stated in the methods. Gene expression is labile so the reader should know if other variables influence the results presented or not.

Thank you for suggestion. We have added detailed information (i.e., age, collecting time and season) to the supplementary Table 1. We have also added this information to the Methods. Because the birds used in transcriptomic analysis (Supplementary Table 1) were different individuals from those used in the sequence analyses (Supplementary Table 2), these two tables cannot be merged.

References:

Campbell-Staton SC, Velotta JP, Winchell KM. 2021. Selection on adaptive and maladaptive genes expression plasticity during thermal adaptation to urban heat islands. Nat. Commun. 12: 6195.

Ghalambor CK, Hoke KL, Ruell EW, Fischer EK, Reznick DN, Hughes KA. 2015. Non-adaptive plasticity potentiates rapid adaptive evolution of gene expression in nature. Nature 525:372–375.

Hao et al. 2023. Divergent contributions of coding and noncoding sequences to initial high-altitude adaptation in passerine birds endemic to the Qinghai–Tibet Plateau. Mol. Ecol. Doi: 10.1111/mec.16942.

Ho WC, Zhang J. 2018. Evolutionary adaptations to new environments generally reverse plastic phenotypic changes. Nat. Commun. 9: 350.

Ho WC, Zhang J. 2019. Genetic gene expression changes during environmental adaptations tend to reverse plastic changes even after correction for statistical nonindependence. Mol. Biol. Evol. 36:604–612.

Ho WC, Li D, Zhu Q, Zhang J. 2020. Phenotypic plasticity as a long-term memory easing readaptations to ancestral environments. Sci. Adv. 6: eaba3388.

Kuo KC, Yao CT, Liao BY, Weng MP, Dong F, Hsu YC, Hung CM. 2023. Weak gene-gene interaction facilitates the evolution of gene expression plasticity. BMC Biol. 21: 57.